# Influence of Joint Flexibility, Hand Grip Strength and Pain on Oral Hygiene in Musculoskeletal Disorders—A Non-Interventional Clinical Study

**DOI:** 10.3390/jcm12062190

**Published:** 2023-03-11

**Authors:** Irshad Ahmad, Rafi Ahmad Togoo, Demah Saleh M. Alharthi, Atheer Ali M. Alhassan, Deena Ali O. Alqahtani, Debjani Mukherjee, Mohammed Saleh Alharthi, Fuzail Ahmad, Hashim Ahmed, Shahnaz Hasan, Mohammed M. Alshehri, Shadab Uddin

**Affiliations:** 1Department of Medical Rehabilitation Sciences, College of Applied Medical Sciences, King Khalid University, Abha 61413, Saudi Arabia; 2Department of Pediatric Dentistry & Orthodontic Sciences, College of Dentistry, King Khalid University, Abha 61413, Saudi Arabia; 3Dental Interns, College of Dentistry, King Khalid University, Abha 61413, Saudi Arabia; 4Department of Physical Therapy, King Fahad Specialist Hospital, Dammam 31444, Saudi Arabia; 5College of Applied Sciences, Almaarefa University, Riyadh 13713, Saudi Arabia; 6Department of Medical Rehabilitation Sciences, College of Applied Medical Science, Najran University, Najran 55461, Saudi Arabia; 7Department of Physical Therapy and Health Rehabilitation, College of Applied Medical Sciences, Majmaah University, Majmaah 15341, Saudi Arabia; 8Department of Physical Therapy, Faculty of Applied Medical Sciences, Jazan University, Jazan 45142, Saudi Arabia

**Keywords:** oral hygiene, joint pain, joint movements, professionals

## Abstract

Diseases of the oral cavity and musculoskeletal disorders (MSDs) are common occurrences. They are commonly linked with partial mobility, resulting in limited visits to dentists for routine oral care, causing poor periodontal condition, bleeding, gingival inflammation, and increased depth of periodontal pockets. The present study was conducted to measure joint movements, hand grip strength, and pain in joints and their association with oral health. Materials and methods: The study included 200 subjects, half suffering from back, neck, shoulder, elbow, and wrist problems, belonging to different age groups and professions; 100 had joint pain, and 100 were without joint pain. The study assessed individuals exposed to oral health issues by measuring the strength of hand grip, flexibility, and pain score of the back, neck, shoulder, elbow, and wrist. The strength of the hand grip and the angle of the elbow and shoulder were measured in addition to a questionnaire to assess the relationship between back pain and oral health. To evaluate dental health status and oral hygiene, the total number of decayed, missing, and filled teeth (DMF/T) and Oral Hygiene Index—Simplified (OHI-S) indices were used. Results: The difference between all demographic parameters was statistically significant (*p* < 0.05). It was observed that there was a significant difference in calculus, debris, and dental caries scores in both groups, with significantly lower scores and better oral hygiene in patients without joint pain. Concerning pain score and joint movements, the group without joint pain showed a significantly better range of movements and less pain than patients suffering from joint pain, and statistically a significant difference (*p* < 0.05) was observed between both groups. Conclusion: The present study revealed that musculoskeletal disorders, pain in the neck and hand, and restricted movements were common among professionals. We observed that pain in joints, neck, and hands, with restricted movements, caused a serious impact on the maintenance of oral hygiene practices among subjects of different professions.

## 1. Introduction

Oral hygiene is a practice to keep the mouth, teeth, and gums clean and far away from many diseases. Maintaining good oral hygiene means the prevention of diseases to come in the future because of poor oral hygiene. Proper brushing techniques, brushing twice, flossing and the use of mouthwash are some of the recommended practices to maintain oral hygiene. Oral cavity diseases and musculoskeletal disorders (MSDs) are common occurrences. These diseases are severe disorders affecting persons of different age groups. However, little consideration has been given to the interrelation between both the disorders. Various musculoskeletal disorders, such as Paget’s disease, osteoporosis, and arthritic disorders affecting people of old age, can directly or indirectly affect their oral cavity and surrounding structures. Several drugs being used to manage musculoskeletal disorders, such as corticosteroids and bisphosphonates, increase the risk of immune system suppression and lead to osteonecrosis of the jaw. People suffering from rheumatoid arthritis, disabling osteoarthritis, and some other conditions find difficulty in managing good oral hygiene and face difficulties in traveling to get professional help [1].

It has been observed that work-related musculoskeletal disorders (WMSDs) have been recognized as an important occupational health issue affecting different professionals [2]. Musculoskeletal disorders are said to be work-related when the performance of work and environment contribute significantly to the disorder. The condition worsens or extends for longer durations because of work issues. Typical work practices that can cause WMSDs include regular overhead work, regular heavy objects lifting, working with the neck in a regular flexed position, the routine experience of whole-body vibration, or doing regular forceful tasks [3].

MSDs are not bound to affect only a particular area of the body, but they usually affect regions such as the lower back, neck, wrists, and shoulders. Patients having MSDs can suffer from tingling, numbness, decreased strength, pain, or swelling of the affected area. These symptoms arise via various mechanisms, such as impaired muscular function, affected conduction of nerves, strains, damage to tendons, muscles and ligaments, or microfractures or degeneration of the bones. Various MSDs are recognized in the medical literature and involve tendonitis, thoracic outlet syndrome, carpal tunnel syndrome, and de-Quervians disease [4,5,6,7,8].

Joint flexibility means when a joint can move freely without any discomfort in an unrestricted range of motion. Muscle weakness and joint pain and musculoskeletal disorders all are interrelated, and can lead to poor quality of life in general and oral hygiene issues specifically. Handgrip strength is an established indicator of better muscular strength. Low handgrip strength is linked to psychological, functional, and social health domains and can be used as an indicator of being healthy. As good oral health reflects the overall wellbeing of an individual, thus hand grip can be used as an indicator for assessing oral hygiene.

Obtaining medical or dental care is a problem for some people with impaired functional status, especially those who are home-bound or reside in long-term care facilities. People with disabling musculoskeletal conditions are likely to be affected by poor oral hygiene [5].

Average brushing time is around 45 s and is done once a day by the general population. In the case of musculoskeletal disorders, joints restriction, joint pain, and hand muscle weakness, these can further reduce the quality and brushing time of an individual, leading his/her poor oral hygiene. Brushing twice for around 2 min in the morning and 2 min evening with 10 strokes per section of each quadrant is recommended for better oral hygiene [9]. We hypothesize that joint restriction, muscles weakness and joint pain in the upper limbs affect the quality of oral hygiene.

The present study was conducted to measure joint movements, hand grip strength, and pain in joints and their association with oral health.

## 2. Materials and Methods

The study was conducted from April 2021 to June 2022. The present case–control study was conducted in the Department of Physical Therapy and College of Dentistry at King Khalid University, Abha, Kingdom of Saudi Arabia. The study was conducted following the Declaration of Helsinki and was approved by the Institutional Ethics Committee (IRB/KKUCOD/ETH/2020-21/060). Initially, 225 male and female subjects were recruited who were with or without joint pain; 25 subjects were excluded because of not fulfilling the inclusion criteria, which include not attending the physical therapy department, surgery in the upper limbs and dental scaling. Finally, the study included 200 subjects, 100 with joint pain and 100 without joint pain, belonging to different age groups and professions. Sample size calculation was performed using a formula (Equation (1)). All the subjects were evaluated by Physical Therapists (having more than 20 years of musculoskeletal clinical experience) for their joints’ range of motion, hand grip strength, and pain intensity. The same subjects were evaluated by a Senior Clinical Dentist for their oral health check-up (Figure 1).
(1)$$N=\frac{Z^{2}\;P\;Q}{e^{2}}$$

*N* = Sample Size;

*Z* = Confidence Level (1.96);

*P* = Prevalence Level of Musculoskeletal Disorders (0.15);

*Q* = 1 – P = 1 – 0.15 = 0.85;

*e* = Error Term = 0.05;

$N=\frac{1.96^{2}*0.15*0.85}{0.05^{2}}$ = 196 (we have taken the round figure 200 for the final study and divided equally into two groups).

### 2.1. Procedure

#### 2.1.1. Joint Range of Motion Measurement

The joint range of motion for neck, shoulder, elbow, and wrist was measured using a standard transparent goniometer; this goniometer has two arms, a moving arm, a stationery arm, and a fulcrum. The fulcrum was placed on the side of a joint; the moving arm was placed along the moving body parts, whereas the stationary arm was placed along the body or a fixed body part. Subjects were asked to move the distal or moving part of the joint, and the physical therapist recorded how many degrees the moving part had moved from 0. There are different types of goniometers available on the market. Due to their low cost and ease of transport, they are widely used in many clinical setups [10,11]. A BASELINE plastic standard goniometer was used in this study (Figure 2).

#### 2.1.2. Hand Grip Strength Measurement

The hand grip strength was measured by using a calibrated handheld Jammer Dynamometer. The patient was asked to sit in an examination chair with their shoulder adducted, elbow flexed to 90 degrees, and arm supported on the armchair, and to place the wrist in a neutral position. Further, the patient was asked to perform the test; the procedure of the test was to hold the grip for 3 s. The test was repeated thrice and the average of three trials was recorded for both hands and each trial had 60 s rest in each trial [12] (Figure 3).

#### 2.1.3. Pain Intensity Measurement

Pain intensity was measured by Visual Analogue Scale (VAS) on a 10 cm vertical number line; one of the ends starts with 0 and the other ends with 10, where 0 represents no pain and 10 indicates the worst pain. VAS is a very commonly used pain measurement tool in musculoskeletal rehabilitation in pre and post treatment outcomes, and this measurement tool is considered valid and reliable [13].

#### 2.1.4. Health Awareness Questionnaire

A set of questionnaires was distributed to the patients to assess the relationship between joint pain and oral health. Later, the validity of the questionnaire was assessed and found to be appropriate (α = 0.85).

### 2.2. Oral Hygiene Examination

#### 2.2.1. Evaluation of Dental Health and Oral Hygiene

To evaluate dental health status and oral hygiene, the total number of decayed, missing, and filled teeth (DMF/T) was examined. Furthermore, Oral Hygiene Index–Simplified (OHI-S) indices were used. The intra-oral examination was conducted according to the World Health Organization’s standards, and caries were assessed by using the DMFT index. Information on the Decayed (D), Missing (M), and Filled (F) teeth due to dental caries were charted on the oral examination assessment form. The OHI-S had two components, the Debris Index and the Calculus Index. Each of these indexes, in turn, was based on numerical determinations representing the quantity of debris or calculus found on the pre-selected tooth surfaces. The following six surface criteria for OHI-S were selected from four posterior and two anterior teeth [14].

Criteria for classifying debris:No debris or stain present;Soft debris covering not more than one third of the tooth surface, or presence of extrinsic stains without other debris regardless of surface area covered;Soft debris covering more than one third but not more than two thirds of the exposed tooth surface;Soft debris covering more than two-thirds of the exposed tooth surface;

Criteria for classifying calculus:5.No calculus present;6.Supragingival calculus covering not more than a third of the exposed tooth surface;7.Supragingival calculus covering more than one third but not more than two thirds of the exposed tooth surface, or the presence of individual flecks of subgingival calculus around the cervical portion of the tooth, or both;8.Supragingival calculus covering more than two thirds of the exposed tooth surface, or a continuous hard band of subgingival calculus around the cervical portion of the tooth, or both.

#### 2.2.2. Calculation Example

After recording the scores for debris and calculus, the index values were calculated. The debris scores were summed and divided by the number of surfaces scored for each individual. At least two of the six possible surfaces were examined for an individual score to be calculated. After that, the score for a group of individuals was obtained by computing the average of the individual scores. The same methods were used to obtain the calculus scores or the Simplified Calculus Index (CI-S). The average individual or group debris and calculus scores were combined to obtain the Simplified Oral Hygiene Index [15].

Inclusion: The age group of the participants in the study was 19 years minimum and upwards. The study included male and female subjects, who participated willingly for oral hygiene screening and physical examination for joint range of motion measurement, hand grip strength, pain assessment. Exclusion: Individuals suffering from ailments such as arm, wrist, or any finger amputation, dominant hand (brushing hand) paralysis, recent injury in their brushing hand, using any splint, cast or bandage, or surgery on their brushing hands.

### 2.3. Data Analysis

The data collection was carried out with statistical analysis by using IBM SPSS 20.0 version. Descriptive statistics such as distribution (frequency and percentage) were used, central tendency measure (mean and standard deviation) was used to describe the quantitative and categorical variables regarding percentage distribution, and mean value was compared between the groups. Student’s *t*-test was used for independent samples to compare the mean values of quantitative outcome variables.

The data were also assessed based on demographic characteristics such as gender, profession, and location of the pain. The patients in both the groups were evaluated for various oral hygiene parameters, so as to observe the effect of joint pain on oral hygiene.

## 3. Results

In the current study, the subjects in the joint pain group suffered daily pain episodes in their brushing hand. They believed that the quality and time of brushing their teeth were significantly affected due to joint pain caused by the brushing. Further, the frequency of brushing decreased significantly among the patients as a result of pain (*p* < 0.05) (Table 1). In both the groups, homemakers, commonly known as housewives, had the highest percentage of 34% in the joint pain group and 28% in the without joint pain group, followed by teachers, 20% (Table 2). The patients with joint pain mostly suffered from shoulder pain, 22%, followed by lumbar, 16%, and cervical pain, 14%. The difference between all demographic parameters was found to be statistically significant (*p* < 0.05) (Table 2).

In both the groups, there were no mean differences in the age of the participants. It was observed that there was a significant difference in calculus, debris, and dental caries scores in both groups, with significantly lower scores and better oral hygiene in patients without joint pain. Statistically, a significant difference (*p* < 0.05) was observed between both the groups in relation to DMFT score 19.68, calculus score 0.5526, and debris index score 1.5010. (Table 3); they were quite different from the other group. Concerning pain score and joint movements, the group without joint pain showed a significantly better range of movement than the patients suffering from the joint pain, and statistically a significant difference (*p* < 0.05) was observed between both the groups (Table 4). It was seen that subjects without joint pain showed better hand grip strength than subjects with joint pain; statistically, a significant difference (*p* < 0.05) was observed between both the groups in relation to the hand grip strength (Table 4).

## 4. Discussion

Oral hygiene practices are based on regular practices, including brushing and flossing, to keep the mouth healthy and disease free, thus keeping the prevalence of dental caries and periodontal diseases low. The impairment of oral hygiene can happen because of the mutilation of functions due to pain, leading to disturbance in daily activities, reduced working hours, financial loss, and impact on education and various other social activities. These disturbances impact the quality of life, thus decreasing the overall efficiency of an individual and raising the financial burden on the family and society [16,17].

Musculoskeletal disorders and diseases of the oral cavity are the most prevalent disabling disorders affecting individuals of all ages and belonging to different professions [18,19,20,21]. In this area, very limited research has been conducted to show the relationship between joint restriction, hand muscle strength, and joint pain. Nevertheless, a few studies reveal the impact of musculoskeletal disorders on oral hygiene. In one of the studies, the authors observed that people who have musculoskeletal disorders such as rheumatoid arthritis, osteoarthritis, and other conditions show difficulty practicing good oral hygiene measures, and they even have issues being treated by dental professionals. In the joint pain group, we noticed that the shoulder, lumbar and cervical joints were more involved; this may affect the patient’s functional activities, leading to poor oral hygiene and fewer dental clinic visits; similar results were found in a study carried out to see the influence of musculoskeletal conditions on oral health [1].

Thus, the present study was conducted to find a relationship between restricted, painful joint and hand grip strength and oral hygiene practices. The finding of our study is that pain in joints, restricted movements, and hand grip weakness causes poor oral hygiene among professionals of various age groups. In the joint pain group, the pain affected the range of joint movements that led to hand grip weakness, and patients were unable to maintain good oral hygiene status as compared to the other group.

The participants showed a reduction in cervical flexion, lumbar flexion, shoulder internal rotation, and elbow flexion with pronation range of motion compared to other groups. These movements play an essential role in brushing teeth as brushing techniques involve combined and coordinated movements of the upper extremities.

It has been advocated that compromised oral hygiene in professionals suffering from restricted movements and joint pain can be managed using various simple and inexpensive methods, such as using high-fluoride toothpastes and antimicrobial mouthwashes. When mechanical measures are not possible, it is recommended to use chlorhexidine mouthwashes to decrease plaque accumulation [22]. Electric toothbrushes are used and helpful in maintaining better oral hygiene without any professional guidance. For reducing plaque and decreasing gingival bleeding, using sonic and ultrasonic toothbrushes is advantageous [23].

In our study, as we have observed, participants were mostly homemakers; this may be due to their awareness of dental issues. Studies have reported that women are more concerned and care more about oral hygiene [24].

Some studies found a correlation between dental occlusion and the physical fitness of individuals. Handgrip strength is a measurement score used to assess internal strength [25]. In a study carried out on adults for oral health, the findings are that oral health is linked with handgrip strength. Poor oral self-care habits can be the risk indicator for low muscular strength [26]. One of the studies revealed a significant correlation between a higher number of teeth and absolute handgrip strength [27].

A number of studies found an association between tooth loss and relative handgrip strength [28]. The result of one study shows an association between oral hygiene maintenance and low muscular strength [29]. A study result found that handgrip strength is an established indicator of the condition of muscular strength, mainly in older individuals [30].

The supposed mechanisms concerning the association between oral health and handgrip strength are varied and remain unclear [31]. According to a study, consuming branched-chain amino acids is linked with handgrip strength. Thus, a direct or indirect effect of tooth loss on physical strength can describe this link with handgrip strength to an assured degree [32]. Studies show that increased levels of inflammatory factors affecting muscle strength were seen in patients with periodontal disease [33]. Thus, all these studies advocate the relationship of handgrip strength to oral hygiene in patients, and the findings of these studies support our results.

In relation to pain scoring, the participants with joint pain were more affected in the lumbar spine, followed by cervical and shoulder joints. There is a direct relation between impaired physical functions of upper limbs and oral health [26]. There have been few studies carried out to see the effects of pain in musculoskeletal joints on oral hygiene, but to date no studies have been undertaken to see which joint pain affects oral hygiene more. Therefore, it will be difficult to generalize our findings and say that if high pain score in lumbar, cervical and shoulder joints are present then there is a higher chance of poor oral hygiene. Since our study was on a specific population, absolute generalizability was not possible, but the help of the correct sample size, right sampling technique and good intervention helped us in the better generalization of the study results. This is a unique study, to our best of knowledge, in Saudi Arabia; this area has not been discussed, although it is a major issue patients are suffering from, and it is an innovative and novel study.

Future studies need to be carried out to check which of the indicators, such as joint range of motion, hand grip muscle strength or joint pain severity, affects oral hygiene collectively. Furthermore, a study could be conducted to see the effects of pre- and post-physical therapy intervention in joint pain subjects on oral hygiene.

Study limitations: The present study is one of a unique kind to date. Nevertheless, we faced some limitations; the study was conducted with a limited sample size, thus further studies should be conducted with a large sample size. Further studies should be conducted to evaluate the awareness and knowledge of different professionals regarding the effect of the condition of joints on oral hygiene.

## 5. Conclusions

The present study revealed that musculoskeletal disorders, pain in the neck, back, shoulders, and hand joints, restricted movements, and hand grip weakness were common among professionals. Due to this, joint pain, restriction of movement, and hand grip weakness caused hindrance in oral hygiene practices among participants coming from different professions backgrounds, increasing the incidence of dental caries, calculus, and poor oral hygiene. Thus, it is important to implement various awareness programs among people of different professions regarding the association between joint pain and dental issues. Further, it is recommended to diagnose and treat these disorders affecting muscles and joints at the earliest, so that good oral hygiene can be established and various dental diseases are prevented.

## Figures and Tables

**Figure 1 jcm-12-02190-f001:**
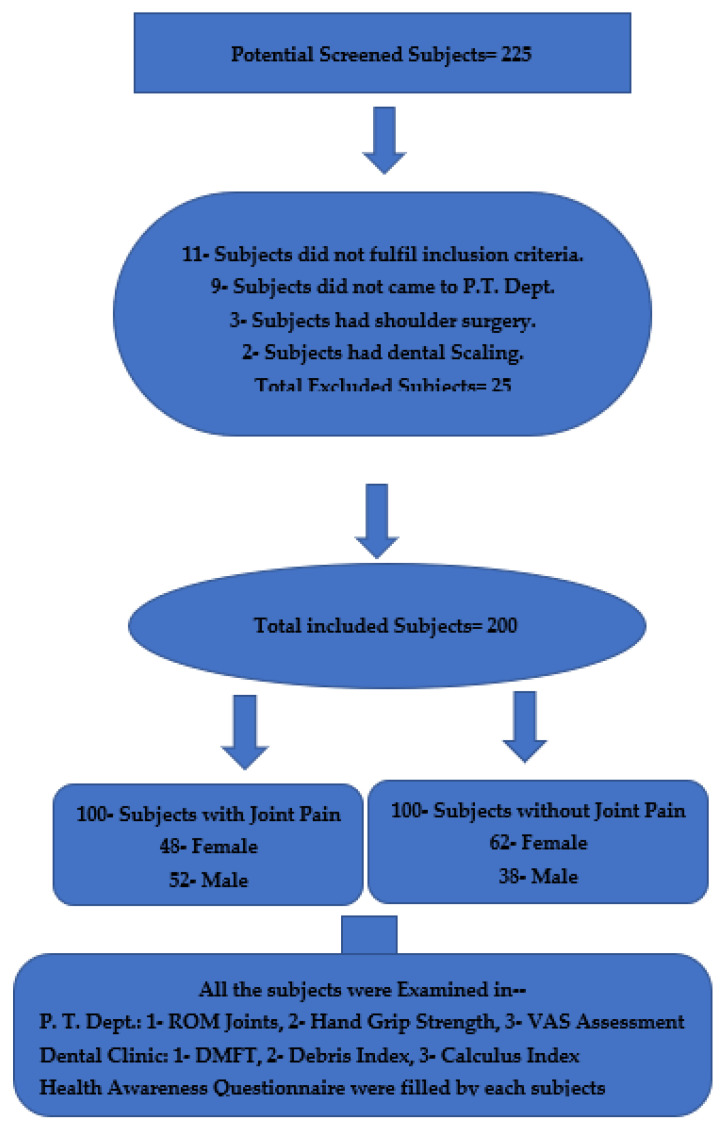
Participants’ flowchart.

**Figure 2 jcm-12-02190-f002:**
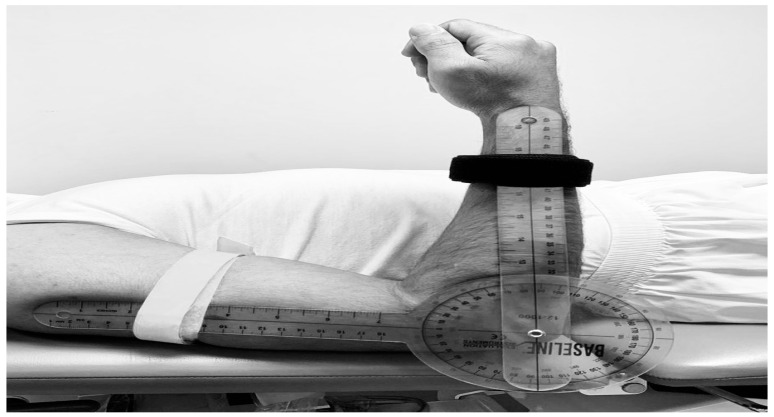
Range of motion measurement with goniometer.

**Figure 3 jcm-12-02190-f003:**
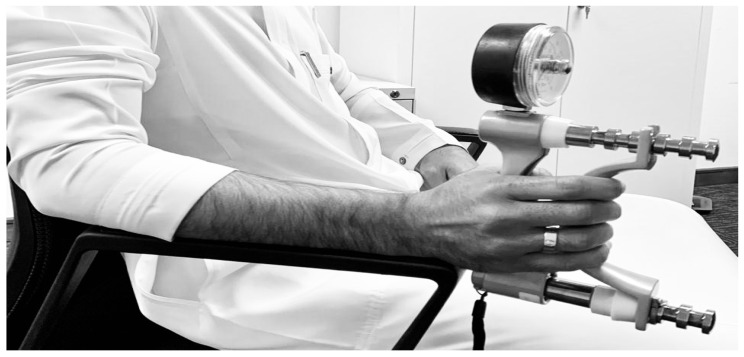
Handgrip strength measurement with handheld Jammer Dynamometer.

**Table 1 jcm-12-02190-t001:** Distribution of study subjects in both the groups according to the questionnaire.

Questionnaire		With Joint Pain (*n* = 100)		Without Joint Pain (*n* = 100)		Statistical Analysis	
Question	Options	Frequency	Percentage	Frequency	Percentage	Chi Square	*p*-Value
How often do you brush?	Never	0	0.0%	4	4.0%	1.787	0.025 *
	Once a day	22	22.0%	32	32.0%		
	Twice a day	68	68.0%	50	50.0%		
	Thrice a day	10	10.0%	14	14.0%		
Pain effects oral health negatively	Maybe	16	16.0%	18	18.0%	2.889	0.012 *
	No	60	60.0%	52	52.0%		
	Yes	24	24.0%	36	36.0%		
Does the brushing time decrease with pain?	Maybe	8	8.0%	2	2.0%	0.120	0.211
	No	46	46.0%	56	56.0%		
	Yes	46	46.0%	42	42.0%		
Does the quality decrease with pain?	Maybe	6	6.0%	8	8.0%	0.109	0.009 *
	No	36	36.0%	72	72.0%		
	Yes	58	58.0%	20	20.0%		

* *p* < 0.05 is significant.

**Table 2 jcm-12-02190-t002:** Distribution of study subjects in both the groups according to demographic factors.

Demographic Factors		With Joint Pain		Without Joint Pain		Statistical Analysis	
Parameter	Options	Frequency	Percentage	Frequency	Percentage	Chi Square	*p*-Value
Sex	Female	48	48.0%	62	62.0%	1.015	0.002 *
	Male	52	52.0%	38	38.0%		
Profession	Carpenter	0	0.0%	0	0.0%	1.290	0.011 *
	Doctor	10	10.0%	4	4.0%		
	Hairdresser	2	2.0%	0	0.0%		
	Housewife	34	34.0%	28	28.0%		
	Manager	4	4.0%	4	4.0%		
	Martial	8	8.0%	8	8.0%		
	Nurse	2	2.0%	0	0.0%		
	Physiotherapist	2	2.0%	2	2.0%		
	Retired	2	2.0%	2	2.0%		
	Student	2	2.0%	18	18.0%		
	Receptionist	2	2.0%	2	2.0%		
	Other	2	2.0%	12	12.0%		
	Teacher	30	30.0%	20	20.0%		
Location of pain	Cervical	14	14.0%	No joint pain		-	-
	Cervical and lumbar	6	6.0%				
	Elbow and shoulder and cervical and lumbar	4	4.0%				
	Lumbar	16	16.0%				
	Shoulder	22	22.0%				
	Shoulder and cervical and lumbar	12	12.0%				
	Shoulder and lumbar	6	6.0%				
	Wrist and elbow and lumbar	4	4.0%				
	Wrist and elbow and shoulder	2	2.0%				
	Wrist and elbow and shoulder and cervical	4	4.0%				
	Wrist and elbow and shoulder	2	2.0%				
	Wrist and elbow and shoulder and lumbar	4	4.0%				

* *p* < 0.05 is significant.

**Table 3 jcm-12-02190-t003:** Mean values of oral hygiene parameters in both the groups.

Parameters	With Joint Pain		Without Joint Pain		Statistical Analysis	
	MEAN	SD	MEAN	SD	*t*-Test	*p*-Value
Age	42.400	11.37308	41.040	11.59514	2.109	0.077
Calculus	0.5526	0.61897	0.3970	1.69553	0.091	0.032 *
Debris	1.5010	0.78464	0.6306	0.44452	3.101	0.024 *
DMFT	19.6800	3.54240	16.7800	3.45992	0.990	0.001 *

* *p* < 0.05 is significant.

**Table 4 jcm-12-02190-t004:** Mean values of ROM, hand grip strength & VAS, parameters in both the groups.

Parameters	With Joint Pain		Without Joint Pain		Statistical Analysis	
	MEAN	SD	MEAN	SD	*t*-Test	*p*-Value
Sh_Fex	163.74	14.196	180.00	0.000	2.09	0.021 *
Sh_Ext	42.90	4.339	50.00	0.000	1.119	0.033 *
Sh_Abd	162.26	19.171	180.00	0.000	1.980	0.045 *
Sh_IntRot	42.26	4.647	55.00	0.000	1.089	0.022 *
El_Flx	125.4800	51.50163	135.0000	0.00000	2.008	0.048 *
El_Pro	78.7000	6.84836	90.0000	0.00000	1.890	0.036 *
Wr_Flx	72.2727	4.67099	80.0000	0.00000	0.456	0.051 *
Cr_Flx	43.56	4.475	60.00	0.000	1.341	0.04 *
Cr_Ext	38.94	7.870	75.00	0.000	2.781	0.002 *
Cr_Rt/Lt-Flx	44.28	12.198	45.00	0.000	1.781	1.000
Cr_Rt/Lt-Rot	57.00	16.684	80.00	0.000	2.655	0.019 *
Lu_Flx	28.9000	27.15131	-	-	-	-
Lu_Ext	21.3704	5.69175	25.0000	0.00000	1.671	0.098
HandGrip Rt	30.0800	7.56965	40.0000	0.00000	2.599	0.039 *
HandGrip Lt	30.3200	8.42116	38.0000	0.00000	2.561	0.027 *
VAS_Cr	7.13	1.455	0.00	0.000	1.334	0.003 *
VAS_Sh	6.60	1.632	0.00	0.000	1.617	0.004 *
VAS_El	1.1200	2.36160	0.0000	0.00000	1.561	0.002 *
VAS_Wr	6.0909	0.94388	0.0000	0.00000	1.590	0.018 *
VAS_Lu	7.5556	1.39596	0.0000	0.00000	2.058	0.029 *

* *p* < 0.05 is significant. Sh—shoulder, Fex—Flexion, Ext—Extension, Abd—Abduction, IntRot—Internal rotation, El—Elbow, Pro—Pronation, Wr—Wrist, Cr—Cervical, Rt/Lt—Right/Left, Lu—Lumbar.

## Data Availability

All data generated or analyzed during this study are available from first author Irshad Ahmad, and will be provided on request.

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
