# Peer review of "Influence of Joint Flexibility, Hand Grip Strength and Pain on Oral Hygiene in Musculoskeletal Disorders—A Non-Interventional Clinical Study"

_jcm, 2023, doi:10.3390/jcm12062190_

Round 1

Reviewer 1 Report

Dear authors,

The abstract should be more focused. The conclusion should be more concrete. 

Please provide how you calculated the needed number of persons to treat 

The inclusion and exclusion criteria should be further detailed

What is the originality and the novelty of the paper? 

The conclusion should be more focused and sustained by the results

Author Response

Dear sir, All the comments and suggestions were really great that has changed our manuscript in a good shape, we thank for that. We have done all the required changes in our revised manuscript.    

Reviewer 2 Report

Dear Authors,

The article: 'Influence of joints flexibility, hand grip strength and pain on oral hygiene in musculoskeletal disorders- A non-interventional clinical study' was conducted to measure joint movements, hand grip strength, and pain in joints and its association with oral health. 

English language and style should be corrected.

Punctuation mistakes should be corrected, e.g., p value should be in italics.

Introduction, materials and methods are clearly written.

The study group, inclusion and exclusion criteria were discussed in detail. Great presentation of research methods. Very well described statistical tests. Legible graph - figure 1.

It seems to me that the following indicators should also be included as indicators of oral hygiene precision: API (Aproximal Plaque Index), BOP (bleeding on probing), PD (pocket depht). Only the use of the DMF indicator [Decayed (D), Missing (M), and Filled (F) teeth] and the OHI-S is a major simplification. It would be worth expanding later research with more precise tools.

line 155 skip to next page

Add a table with abbreviations.

References should be prepared in accordance with the MDPI guidelines.

To sum up, article should be reconsidered after minor revision.

Author Response

Sir, we have modified our manuscript according to your comments and suggestions. Thanks a lot for your efforts. 

Reviewer 3 Report

- This is an interesting study that analyzed the difference in oral hygiene level according to joint flexibility, hand grip strength and pain in musculoskeletal disorders. However, as a scientific research, some revision is required.

- In the abstract section, “Data collected was subjected to statistical analysis using IBM SPSS 20.0 version.” Please delete it.

- Please revise all ‘p-value<0.05’ in the manuscript to ‘p < 0.05’.

- In the results section, please revise ‘(Table no. 1)’ to ‘ (Table 1) (all tables)

- There is an error in percent calculation in Tables 3.1 and 3.2. Please recalculate as a percentage.

- In 'Table 3.3', there is a significant difference in 'age' between the experimental group and the control group. Please consult a statistician.

- In the discussion section, please add future research directions that can supplement the limitations of the study.

Author Response

Dear sir, we are really happy to see your valuable comments, suggestions and motivations, it helped us a lot for improving our manuscript. Sir we have did all the changes as suggested by you. 

Reviewer 4 Report

This study targets a very challenging subject. It aims to assess joint movements, handgrip strength, and joint pain as well as and its association with dental health and oral hygiene.

I think that the authors did a good job. The manuscript is well written and concise.

However, I have several comments to add:

Data analysis section:

I suggest improving this part and adding how the authors handled descriptive analysis and associations.

Results section:

·      This section must only contain results. Some text must be moved to data analysis section, such as: “The data was also assessed based on demographic characteristics like gender, profession, and location of the pain”; “Patients in both groups were evaluated for various oral hygiene parameters to observe the effect of joint pain status on oral hygiene status”.

·      To be clearer, I suggest dividing this section into subparagraphs with subtitles.

·      The results that appear in the tables must be mentioned in the text. Please adjust.

Discussion section:

·      Please start your discussion section by presenting the main findings, which reply to the study aim.

·      Please highlight the strengths of the study, and discuss in more detail the weak points and limitations, particularly biases, and the generalizability of the results.

The conclusion is more a perspective than a conclusion. I would suggest improving that part.

Author Response

Respected Reviewer sir, we have made required and suggested changes in our manuscript. It was a great experience, working on those comments, we really tanksful and appreciate for your comments.

Round 2

Reviewer 3 Report

Dear Authors,

I check the revised contents of the manuscript. There is an error in communication. Please delete ‘ ’ for all ‘p < 0.05’ in the manuscript.

Author Response

Dear reviewer sir we have made the changes in our manuscript as you suggested and uploaded the revised manuscript.

Thanks sir  
